# Variability of lightning hazard over Indian region with respect to ENSO Phases

Avaronthan Veettil Sreenath[1], Sukumarapillai Abhilash[1,2], and Pattathil Vijaykumar[1,2]

[1]Department of Atmospheric Sciences, Cochin University of Science and Technology, Cochin 682016, India
[2]Advanced Department Centre for Atmospheric Radar Research, Cochin University of Science and Technology, Cochin 682022, India

**Correspondence:** S Abhilash (abhimets@gmail.com)

**Abstract.**

The El-Nino Southern Oscillation (ENSO) modulates the Lightning Flash Density (LFD) variability over India during pre-monsoon, monsoon, and post-monsoon seasons. This study intends to shed light on the impact of ENSO phases on the LFD over the Indian subcontinent using the data obtained from Optical Transient Detector (OTD) and Lightning Imaging Sensors (LIS) onboard the TRMM satellite. Results suggest the LFD over Northeast India (NEI) and southern peninsular India (SPI) strengthened (weakened) during the warm (cold) phase of ENSO in the pre-monsoon season. During monsoon season, NNWI (north of northwest India) shows above (below) normal LFD in the cold (warm) ENSO phase. It is striking to note that there are three hotspots of LFD over the Indian land region, which became more prominent during the monsoon seasons of the last decade. A widespread increase of LFD is observed all over India during the warm phase of ENSO in the post-monsoon season. A robust rise in graupel/snow concentration is found during the post-monsoon season over SPI in the ENSO phase, with the lowest fluctuations over the NEI and NNWI regions. The subtropical westerly jet stream is shifted south in association with the warm phase, accompanied by an increase of geopotential height (GPH) all over India for the same period. This exciting remark may explain the indirect influences of ENSO's warm phase on LFD during the post-monsoon season by pushing the mean position of subtropical westerly towards southern latitudes. However, the marked increase of LFD is confined mostly over the NNWI in the cold ENSO phase.

## 1 Introduction

Lightning is a tremendous and inescapable atmospheric hazard that humankind has encountered throughout history (Cooray et al., 2007; Mills et al., 2010). The number of casualties underlines lightning hazards as a devastating phenomenon, with an annual death rate of 2234 from 2001 to 2014 over India (Selvi and Rajapandian, 2016). Singh and Singh (2015) documented the yearly number of lightning fatalities and lightning flashes in India from 1998 to 2005, and they find that the fatalities increase coherently with the lightning flash rate. Lightning strikes over the plain terrains are observed to be less as compared to the hilly regions. Due to the former's high population density, even lesser lightning flashes take many people's lives due to high chances of being struck by lightning (Yadava et al., 2020).

The El-Nino Southern Oscillation (ENSO) is a naturally occurring planetary-scale phenomenon related to the variations in sea surface temperatures over the tropical Pacific Ocean, strongly influencing the number of flashes and average flash rate (Kumar and Kamra, 2012). It is one of the most dynamic climatic variability modes, characterized by three phases: El-Nino (Warm), La-Nina (Cold), and Neutral. The ENSO is a crucial player in the transport of heat, moisture, and momentum and modulates the frequency, intensity, and location of deep convection and the associated lightning activity (Williams, 1992; Kulkarni and Siingh, 2014). Higher Lightning Flash Density (LFD) areas are located away from the equator during the warm phase and coincide with regions of anomalous jet stream circulation enhanced by the meridional heat transport (Chronis et al., 2008). Kandalgaonkar et al. (2010) has reported that lightning activity during the El-Nino year of 2002 increased by 18% over the Indian land region compared to the La-Nina years during 1998-2011. On a global scale, lightning activity shows strong regional preference during different ENSO Phases.

The changes in the lower and upper air circulations associated with different ENSO phases have been found to influence the storm frequency and intensity (Yang et al., 2002; Hsu and Wallace, 1976), which in turn affect the lightning activity (Goodman et al., 2000). Kent et al. (1995) observed that ENSO could dictate clouds' distribution over the tropics and subtropics. Owing to the presence of anomalous subsidence over the western Pacific and adjacent landmass, deep convective clouds are inhibited; hence the rainfall is less during the warm phase (Cess et al., 2001). A southward/eastward shift in the global lightning activity is visible during the warm phase, and the latitudes corresponding to the descending limb of the Hadley circulation exhibit the most significant contrast of LFD between the warm and cold phase of the ENSO (Sátori et al., 2009).

Generally, lightning activity is tuned by the clouds growing deep into the atmosphere. The deep convective cores present over India's east coast during the pre-monsoon season shift to the foothills of western Himalayas during the monsoon (Romatschke et al., 2010). Cecil et al. (2014) documented that India's offshore regions and the maritime continent are prone to deep convection. The vertical growth of cloud systems is amplified by the intense updraft, promoting ice crystals and super-cooled liquid (mixed-phase) inside the convective system. The interaction between these hydrometeors is mainly responsible for the electrification inside the cloud (Takahashi et al., 1999; Williams, 2001). The atmosphere's dynamic and thermodynamic states also modulate the lightning activity over a region (Williams, 1992; Zipser, 1994; Petersen et al., 1996; Rosenfeld, 1999). Topography is identified as another critical participant in developing deep convective clouds and impacts the distribution of lightning activity (Kilinc and Beringer, 2007). Earlier studies have observed that elevated landmass favours the development of deep convective clouds (Zipser et al., 2006; Houze Jr et al., 2007; Rasmussen and Houze Jr, 2011) and thereby leading to higher LFD. In addition, aerosols are also considered a contributor for making a decisive role in generating lightning flashes. Higher aerosol loading increases the available liquid water in the mixed-phase condition, an essential factor for cloud electrification and lightning activity (Williams et al., 2002). Venevsky (2014) reported a significant correlation between lightning and concentration of annually-averaged cloud condensation nuclei over both land and ocean.

The awareness of lightning safety among the public is relatively low. The present study aims to provide vital information to the public on the risky lightning periods over the Indian subcontinent and how the large scale phenomenon, ENSO, is influencing the same. We are detailing the modulation of LFD under different ENSO phases with the help of a vertical profile of hydrometeors (graupel and snow) inside the cloud systems and related atmospheric dynamics during pre-monsoon (March-

May) monsoon (June-September) and post-monsoon (October-December) seasons in India. The rest of the paper is organized as follows. Section 2 provides descriptions of the data and methodology employed in this study. Section 3 presents the results followed by subsection 3.1, which depicts the composite analysis of LFD for pre-monsoon, monsoon, and post-monsoon seasons corresponding to the three ENSO phases. The remaining subsections of section 3 (3.2, 3.3 and 3.4) portray the composite analysis of anomalous LFD during the different seasons and the significance of vertical cloud structure and associated dynamics in regulating their distribution. Finally, the conclusions of this work are given in section 4.

## 2   Data and methods

The Lightning Imaging Sensor (LIS) was an instrument onboard the Tropical Rainfall Measuring Mission (TRMM) satellite launched in December 1997. This instrument senses lightning flashes across the global tropics and subtropics (Goodman et al., 2007). The Optical Transient Detector (OTD) was the predecessor of LIS, launched in the MicroLab-1 satellite. Combined OTD (Optical Transient Detector) + LIS (Lightning Imaging Sensor) monthly averaged flash density expressed flash $km^{-2}.day^{-1}$ available from http://ghrc.nsstc.nasa.gov/ is used in this work. These products compute mean LFD by accumulating the total number of flashes observed and the entire observation duration for each grid box ($2.5° \times 2.5°$) from the thousands of individual satellite orbits. The lightning climatology derived from OTD / LIS (Cecil et al., 2014) provides a unique observational basis for the global flash distribution in monthly time series (Kamra and Athira, 2016), seasonal cycles (Christian et al., 2003), or diurnal cycles (Blakeslee et al., 2014). To produce the low-resolution monthly time series (LRMTS) data, LIS and OTD flash density and view times are smoothed precisely and are extracted for the middle day of each month (Cecil et al., 2014). The LFD in an LRMTS have slightly over three months of temporal smoothing and $7.5° \times 7.5°$ spatial smoothing (Cecil et al., 2014). The data sets are described in greater detail in the following paper: Gridded lightning climatology from TRMM-LIS and OTD: Dataset description by Cecil et al. (2014).

The LFD data is available starting from July 1995 only. So the pre-monsoon season in our work starts in 1996 (March-May) and ends in 2013 (March-May). Due to data unavailability, the first monsoon season includes only three months (July, August, and September 1995). This particular season terminates in 2013 (June, July, August, and September). On the other hand, the post-monsoon season is prepared from 1995 (October-December) to 2013 (October-December). LFD anomaly in this study indicates the difference between the composite of LFD during a particular ENSO phase in a specific season and the composite of LFD during all the three ENSO phases in that specific season. e.g., LFD anomaly during pre-monsoon during La-Nina = (Composite of LFD during La-Nina in pre-monsoon) - (Composite of LFD during all the three ENSO phases in pre-monsoon). The anomalies of all other parameters used in this study are calculated using the same method.

With the aid of TRMM-3A12 data, the cloud structure is examined by evaluating the vertical profiles of hydrometeors (graupel, snow) and latent heat release during different phases of ENSO. The data set has a spatial resolution of $0.5° \times 0.5°$, available from January 1998 to December 2013. It has 28 vertical levels, which start from 0.5 km, and each level is separated by 0.5 km. These parameters are averaged for the pre-monsoon, monsoon and post-monsoon season from 1998 to 2013 with respect to La-Nina, El-Nino and Neutral phases of ENSO over northeast India (NEI: 85° E-95° E, 20° N-30° N), north of

| Year | Pre-monsoon | Monsoon | Post-monsoon |
|------|-------------|---------|--------------|
| 1995 | 0.39 | -0.08 | *-0.60* |
| 1996 | -0.27 | -0.10 | -0.25 |
| 1997 | **0.51** | **1.81** | **2.41** |
| 1998 | **1.15** | *-0.57* | *-1.20* |
| 1999 | *-0.73* | *-0.77* | *-1.24* |
| 2000 | *-0.80* | -0.39 | *-0.65* |
| 2001 | -0.22 | 0.01 | -0.25 |
| 2002 | 0.27 | **0.81** | **1.40** |
| 2003 | 0.10 | 0.15 | 0.46 |
| 2004 | 0.14 | **0.58** | **0.76** |
| 2005 | 0.41 | 0.14 | -0.36 |
| 2006 | -0.27 | 0.39 | **1.02** |
| 2007 | -0.10 | -0.40 | *-1.40* |
| 2008 | *-0.76* | -0.08 | -0.43 |
| 2009 | -0.35 | **0.73** | **1.49** |
| 2010 | **0.62** | *-0.97* | *-1.52* |
| 2011 | *-0.61* | -0.34 | *-0.96* |
| 2012 | -0.17 | **0.53** | 0.21 |
| 2013 | -0.03 | 0.39 | **0.79** |

**Table 1.** ONI during the pre-monsoon, monsoon, and post-monsoon season in India from 1995 to 2013. The bold, italics and underlined values denote El-Nino, La-Nina, and Neutral phases of ENSO, respectively.

northwest India (NNWI: 25° N-40° N, 65° E-80° E) and southern peninsular India (SPI: 5° N-15° N, 75° E-80° E) and used in this work.

Finally, the modulation of Geopotential height (GPH) at 500 hPa, wind at 200 hPa, and specific humidity (SH) at 300 hPa 95 are also examined with the ENSO phases from July 1995 to December 2013. The above parameters are obtained from the NCEP–National Center for Atmospheric Research (NCAR) reanalysis data with a similar spatial and temporal resolution of LFD. Oceanic Nino Index (ONI) is the standard used to identify different phases of ENSO. The average value of ONI is determined during pre-monsoon, monsoon, and post-monsoon season by using HadISST data and detailed in table 1. If the ONI value is above (below) +0.5° (-0.5°) C, it is taken as the warm (cold) phase, and the neutral phase corresponds to the ONI 100 index lies between -0.5/+0.5° C.

# 3 Results and discussion

## 3.1 Composite LFD with respect to ENSO phases

Figure 1 represents the LFD composites for pre-monsoon, monsoon, and post-monsoon seasons corresponding to the three ENSO phases. Irrespective of ENSO phases, the LFD peak is located over NEI during the pre-monsoon season, while its peak shifts to the NNWI in the monsoon season. Kamra and Athira (2016) identified a higher concentration of LFD over northwest and northeast regions of India, and it is tightly correlated with CAPE over those regions. They also observed that the maxima of lightning during post-monsoon is also lying over India's southern and eastern regions. Ahmad and Ghosh (2017) reported that compared to other Indian regions, lightning activity is higher over the North-Eastern part and southern part of India during the pre-monsoon season. Similarly, we have identified three hot spots of higher lightning activity over the Indian subcontinent (Figure 1 (a)). They are located in the NEI (85°E-95°E, 20°N-30°N), NNWI (25°N-40°N, 65°E-80°E) and SPI (5°N-15°N, 75°E-80°E). The Himalayan orography favours the formation of deep convective systems over the NEI (Goswami et al., 2010) and is evidenced by the high values of LFD over the region. Rather than the altitude, the steep topographic gradient is responsible for producing deep convection. Most likely, the deep convective clouds developed in the conditionally unstable atmosphere during the pre-monsoon season are electrically more active (Williams et al., 1992). Lau et al. (2008) proposed that during the pre-monsoon months, dust and black carbon from neighbouring sources accumulates over the Indo-Gangetic plain against the foothills of the Himalayas and act as an elevated heat pump (EHP). Accordingly, this enhanced warming of the middle and upper troposphere contributes to the genesis of deep clouds and higher LFD.

Compared to monsoon and post-monsoon seasons, convective available potential energy (CAPE) is higher during the pre-monsoon season. The seasonal average of CAPE is highest over India's east coast, and it is near 1500 J/kg all over south India (Murugavel et al., 2014). Nevertheless, large regions of India, especially the central Indian region, show a seasonal average of CAPE less than 1000 J/Kg (Murugavel et al., 2014). Strikingly, the areas of higher values of LFD (Figure 1) during the pre-monsoon season coincide with the regions of CAPE maxima reported by (Murugavel et al., 2014). Previous works ascertain that the moderate updrafts limit the vertical development of convective clouds during the summer monsoon under the influence of maritime air mass (Kumar et al., 2014; Tinmaker et al., 2015), which leads to a decline in the cloud electrification during the monsoon season. Among the three seasons, post-monsoon shows a minimum of LFD over the Indian region (Figure 1). One possible reason for this may be the existence of a low average value of CAPE (<500 J/kg) over most parts of India during this season (Murugavel et al., 2014), which is relatively low to favour the development of deep convection and hence lightning.

The relationship between LFD and graupel concentration is examined during the three seasons by using a Pearson correlation analysis over NEI, NNWI and SPI (Figure 2). It shows that the correlation between LFD and graupel concentration is peaking during the post-monsoon season over NEI (r=0.81), NNWI (r=0.64) and SPI (r=0.62). In contrast, the correlation attained a minimum value during the monsoon season over these hotspots regions of LFD (NEI (r=0.24), NNWI (r=0.36), SPI (r=0.38)). It is important to note that the pre-monsoon season also exhibits solid correlation between LFD and graupel concentration over NEI (r=0.64), NNWI (r=0.42) and SPI (r=0.55). All the correlation values are strong and statistically significant at a 95%

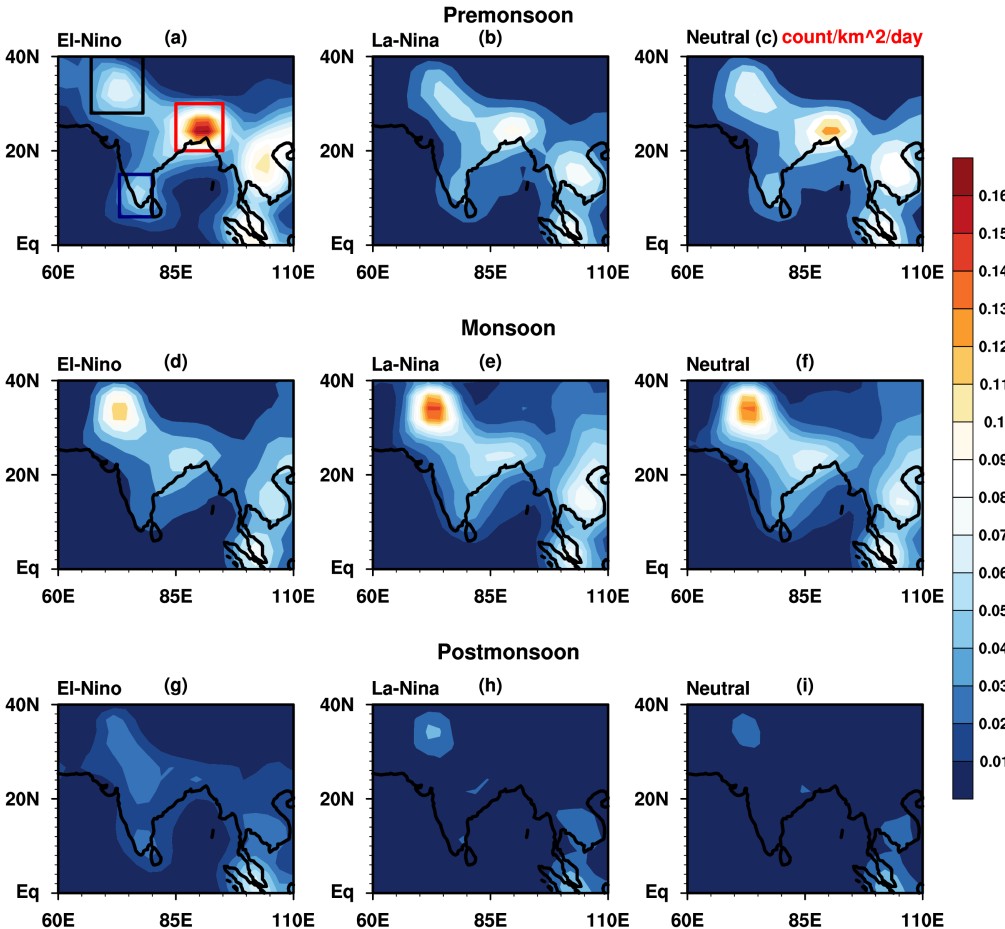

**Figure 1.** LFD composite during different ENSO phases. Coloured boxes in figure 1 a represents the hotspot regions of LFD (red box = NEI, black box = NNWI, blue box = SPI)

confidence level, indicating tight linearity in the relationship of LFD and graupel concentration over regions of higher lightning activity over India.

### 3.2  Distribution of anomalous LFD during pre-monsoon season with respect to ENSO phases

The LFD values are lower than the normal during the pre-monsoon season over NNWI when the ENSO phase is either warm or cold (Figure 3 (a, b)) and it exhibits an increase of LFD during the neutral phase (Figure 3 (c)). While looking into the LFD anomaly of individual years, pre-monsoons of three years (1997, 1998, and 2010) over NNWI have come under the El-Nino phase (Figure 4 (d)). The first two exhibits a decrease of LFD , contributing to the overall reduction in the LFD over NNWI

(Figure 3 (a)). Out of the four La-Nina years (1999, 2000, 2008, and 2011) of pre-monsoon season, 1999, 2000 and 2011 have below-average values of LFD over the same region (Figure 4 (d)).

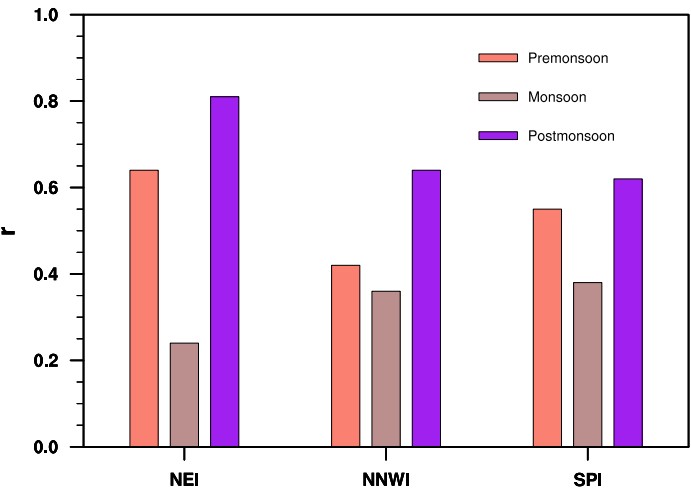

**Figure 2.** Pearson correlation coefficient (r) between LFD and graupel concentration over NEI, NNWI and SPI during pre-monsoon, monsoon and post-monsoon season.

In situ airborne observations during the Cloud-Aerosol Interaction and Precipitation Enhancement Experiment (CAIPEEX) over various locations of India shows that convective clouds during the pre-monsoon and monsoon period have an ice water

content of $10^{-4}$ to 1 $gm^{-3}$ (Patade et al., 2015). Moreover, in situ measured ice cloud properties in the European Cloud Radiation Experiment (EUCREX) have reported a similar range of ice water content inside the clouds system ($10^{-4}$ $gm^{-3}$ to 1 $gm^{-3}$) (Hogan et al., 2006). From TRMM observations and high-resolution model simulations, Abhilash et al. (2008) reported ice concentrations of $10^{-3}$ to $10^{-2}$ $gm^{-3}$ for convective storms over the Indian region. The vertical profiles for graupel and snow concentration are shown over the hotspot domains of LFD in figures 5 and 6, respectively, revealing a significant disparity in

their seasonal average with the observation region. These figures clearly demonstrate that the seasonal average of graupel and snow concentration are peaking around 6 Km over NEI, NNWI and SPI, and after that level, they shows a rapid decrease with height. Note that the seasonal average of latent heat (LH) over these hotspot domains of LFD are peaking between 6-7 Km range and dramatically coincides with the peaking altitude of graupel/snow concentration, mainly because of the release of energy during the phase transition of cloud droplets to ice particles (Figure 7). It is captivating that graupel/snow concentrations are

prominent during all three seasons over SPI, with a maximum value in the post-monsoon season. In contrast to this observation, the NEI and NNWI are showing a minimum value of graupel/snow concentrations during the post-monsoon season in India.

The anomalous profile of graupel in figure 5 (a) indicates that clouds over NEI have high (low) graupel content during the ENSO warm (cold) phase. An increase of snow content with a peak value near 6.5 km is also observed over NEI during pre-monsoon in the warm ENSO phase (Figure 6 (a)). The interaction between snow and graupel and the associated charge

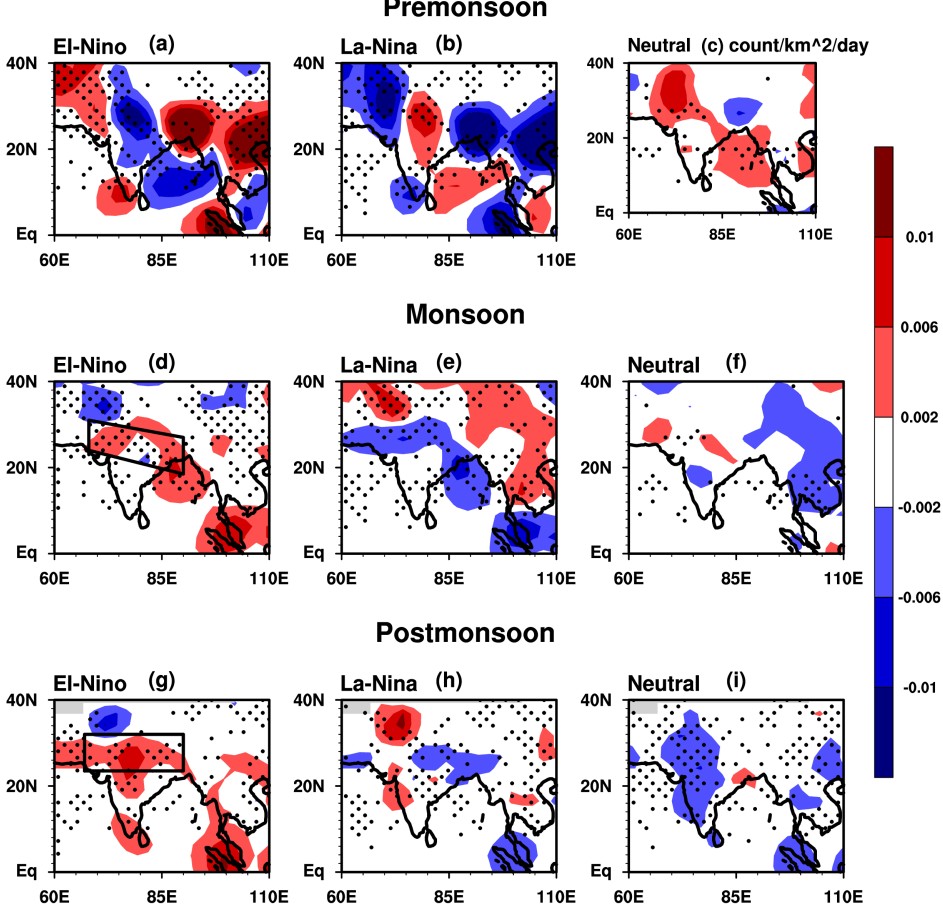

**Figure 3.** Anomaly composite of LFD during different ENSO phases with stippling indicates statistically significant areas at a 95% confidence level. The box in figure 3 (d) and (g) shows the monsoon trough and western disturbance region, respectively.

generation is responsible for lightning from convective clouds. Thus the formation of the higher amount of graupel and snow over NEI during the warm phase will release more latent heat, which is evident from figure 7 (a). Positive (negative) anomalies of SH over NEI indicates that the convective clouds formed during the ENSO warm (cold) phase are vigorous (wimpy) and consequently responsible for the enhanced (reduced) LFD. More importantly, graupel, snow, and latent heat profiles present below-average values in the neutral ENSO phase, which may confirm a decrease in lightning events over NEI.

From figure 7 the anomalous latent heating exists mostly between +/- 0.01. In some cases, it is extending up to +/-0.02 (Kelvin/hr); additionally, previous studies indicate these anomalous values are highly significant (Kumar et al., 2014). Linked with the decrease of graupel and snow content over NNWI during the warm and cold phase of ENSO (Figures 5 (d), 6 (d)),

LFD changes (Figure 7 (d)). The anomalous negative SH at 300 hPa manifests that the clouds are unable to penetrate deep into the atmosphere during these two phases over NNWI (Figure 8 (a, b)). As a result, LFD over NNWI in these phases is low, especially in the cold phase. Contrarily, higher LFD during the neutral phase pointing the abundance of graupel and snow inside the cloud system. It is noticed that ENSO cold phase during the pre-monsoon season is favourable to LFD over central India (CI) (Figure 3 (b)), and the converse is true for ENSO warm phase (Figure 3 (a)).

The graupel and snow concentrations over SPI are anomalously high up to 6 km during the cold phase of ENSO, and above that level, it decreases (Figures 5,6 (d)) . This particular hydrometeor pattern reverses during the warm phase, exhibits below-average values beneath 6 km, and rapidly increases above that level. A similar pattern is observed in the vertical profiles of latent heat release above 6 km. Note that the analysis presented here confirms that the warm phase of ENSO intensifies the deep convection over SPI during the pre-monsoon season and hence promotes LFD.

### 3.3 Distribution of anomalous LFD during monsoon season with respect to ENSO phases

The LFD over the monsoon trough region of India increases during the warm phase of ENSO (Figure 3 (d)) but it remarkably decreases during the cold phase (Figure 3 (e)). Based on the 1998-1999 El-Nino event, Hamid et al. (2001) suggested that intense convective storms developing over the maritime continents are responsible for the increase of lightning activity despite a decrease in the number of convective storms. During the El-Nino years of 1997-1998 and 2002-2003, the southeast Asian regime exhibited an above-average value of lightning (Kumar and Kamra, 2012). While analyzing the 300 hPa SH variability, we noticed that the amount of SH over the monsoon trough is higher during the warm and lower during the cold phase (Figure 8 (d, e)). The NEI is showing a positive anomaly of LFD during the cold phase of ENSO. Figure 4 (b) enforces this result by showcasing that majority of years under the cold phase (during the monsoon season) shows an increase in LFD over NEI. The vertical profile of LH shows an above-average value during the cold phase, and this enhancement of LH in the mid-troposphere helps to increase atmospheric instability and deep convection. On the other hand, the vertical distribution of hydrometers are not displaying any compelling variability with ENSO phases over NEI (Figures 5 (b), 6 (b)). The observed increase (decrease) in the anomalous LFD over NNWI during the cold (warm) period is captured well in the vertical profiles of graupel, snow and latent heat release (Figures 5 (e), 6 (e), 7 (e) ) . There is no noticeable change in the distribution of LFD over SPI in the three phases of ENSO (Figure 3 (d, e, f).

It is interesting to observe that during the 12 years from 2002 to 2013, 11 years have shown above-average values of LFD over the NEI and NNWI regions (Figure 4 (b, e)), registering the intensification of deep convective cloud formation during the recent monsoon season over respective areas. Out of the nine years from 2005 to 2013, 8 have above-normal LFD over SPI (Figure 4 (h)). Thus indicating an escalation of deep convection over SPI in that period. Specifically, the hotspots of LFD over the Indian land region became more prominent during the last decade's monsoon seasons.

### 3.4 Distribution of anomalous LFD during post-monsoon season with respect to ENSO phases

Western disturbances (WD) are the vertical perturbations associated with the subtropical westerly jet stream, which is one of the potential contributors to the rainfall over northern India during the post-monsoon season (Dimri et al., 2016). The jet is

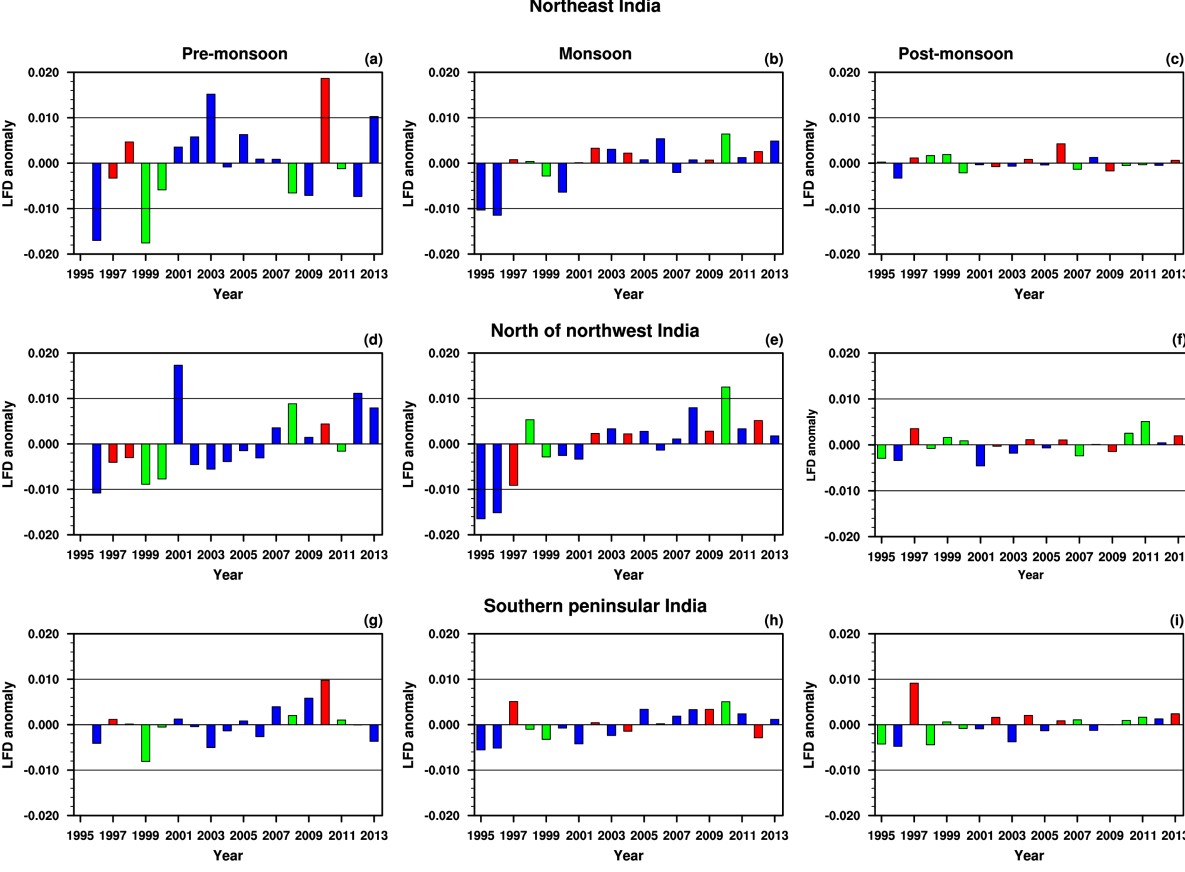

**Figure 4.** Anomalous LFD during the individual years with different ENSO phases. The red colour label bar corresponds to warm, the green one corresponds to cold, and the blue colour label bar indicates the neutral phases of ENSO.

more intense and propagates southward during the El-Nino phase of ENSO (Schiemann et al., 2009). Our analysis shows at the time of the post-monsoon El-Nino period, LFD is increased throughout the country, and it is maximum over north-central India (Figure 3 (g)). In contrast, in the cold phase, intense LFD is concentrated only over the NNWI (Figure 3 (h)). Zubair and Ropelewski (2006) reported a significant role for ENSO in controlling the post-monsoon rainfall over SPI. The SPI is showing
an increase of LFD in the warm phase of ENSO during this season due to the presence of clouds having higher graupel and snow content over that region (Figures 3 (g), 5 (i), 6 (i)). The entire years grouped under the warm phase of ENSO during the post-monsoon season show an increase of LFD over SPI. On the other hand, during the cold phase, anomalous LFD displays an inconsistent pattern of oscillation (Figure 4 (i)).

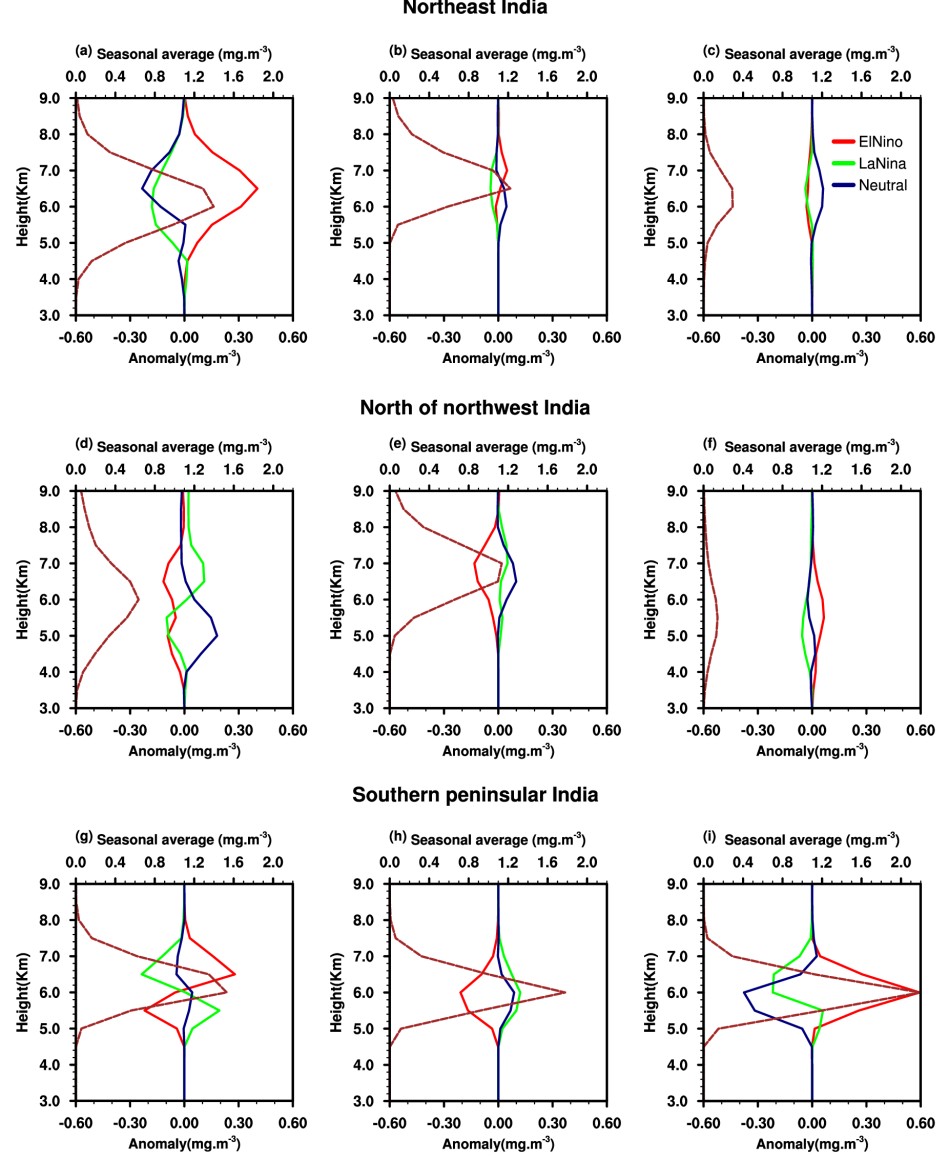

**Figure 5.** Seasonal average (brown dotted curve) and anomaly composite of graupel concentration during different ENSO phases. (a, d, g): pre-monsoon season, (b, e, h): Monsoon season, (c, f, i): post-monsoon season.

Climate variability, like ENSO, can alter the position of jet streams and hence the distribution of WD (Hunt et al., 2018).
Syed et al. (2006) identified that the intensification of WDs during the El-Nino is associated with the weakening of Siberian high. Studies signify that depressions formed over the South Bay of Bengal and the Arabian Sea can also modulate WDs' path

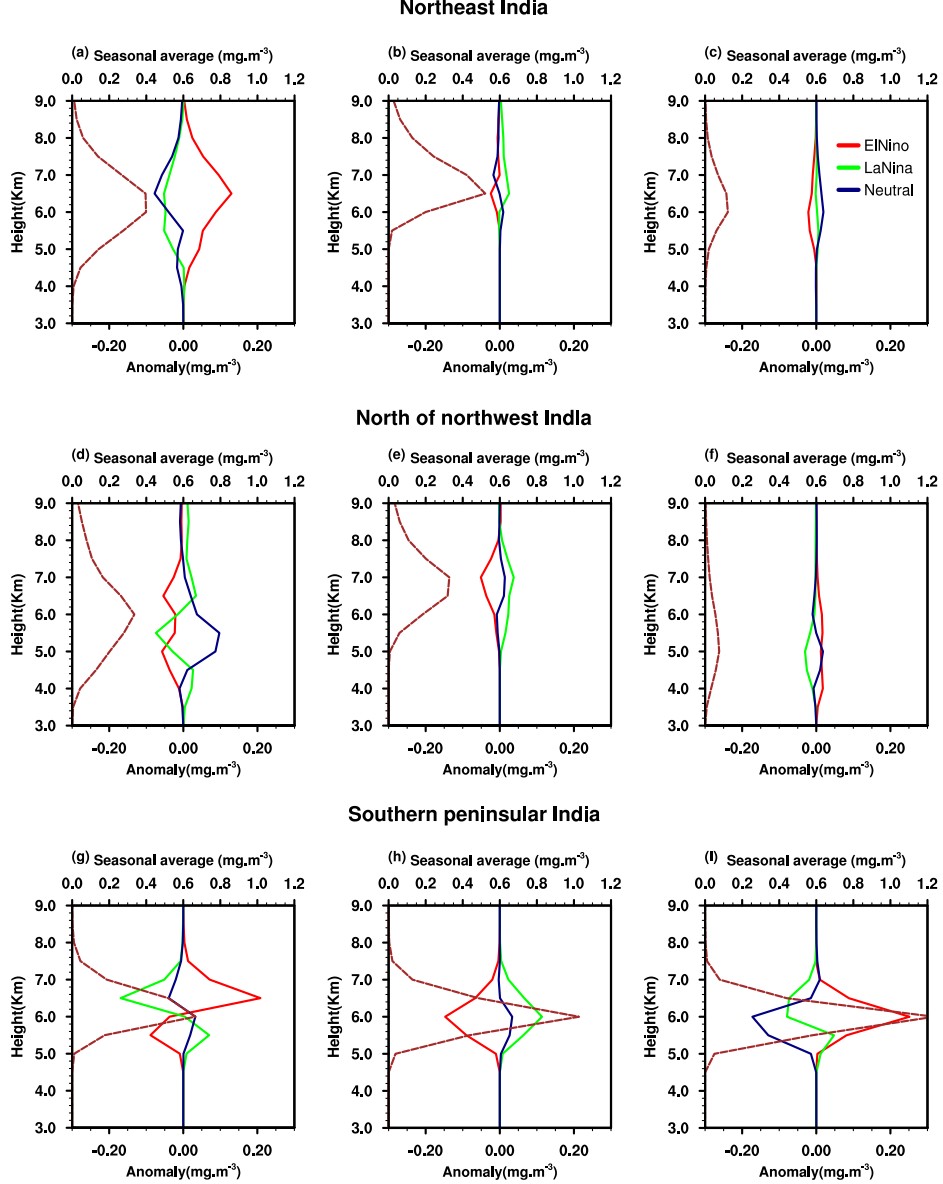

**Figure 6.** Seasonal average (brown dotted curve) and anomaly composite of snow concentration during different ENSO phases. (a, d, g): pre-monsoon season, (b, e, h): Monsoon season, (c, f, i): post-monsoon season.

(Rao et al., 1969). The 500 hPa GP surface drops down (go up) beyond 25°N latitude and indicates the reduction (enhancement) of convection over that region during the warm (cold) phase of ENSO (Figure 9 (a, b)). Meanwhile, a higher (lower) GP surface is visible all over India during the warm (cold) phase, which is an indication of an increase (decrease) in the convective activity

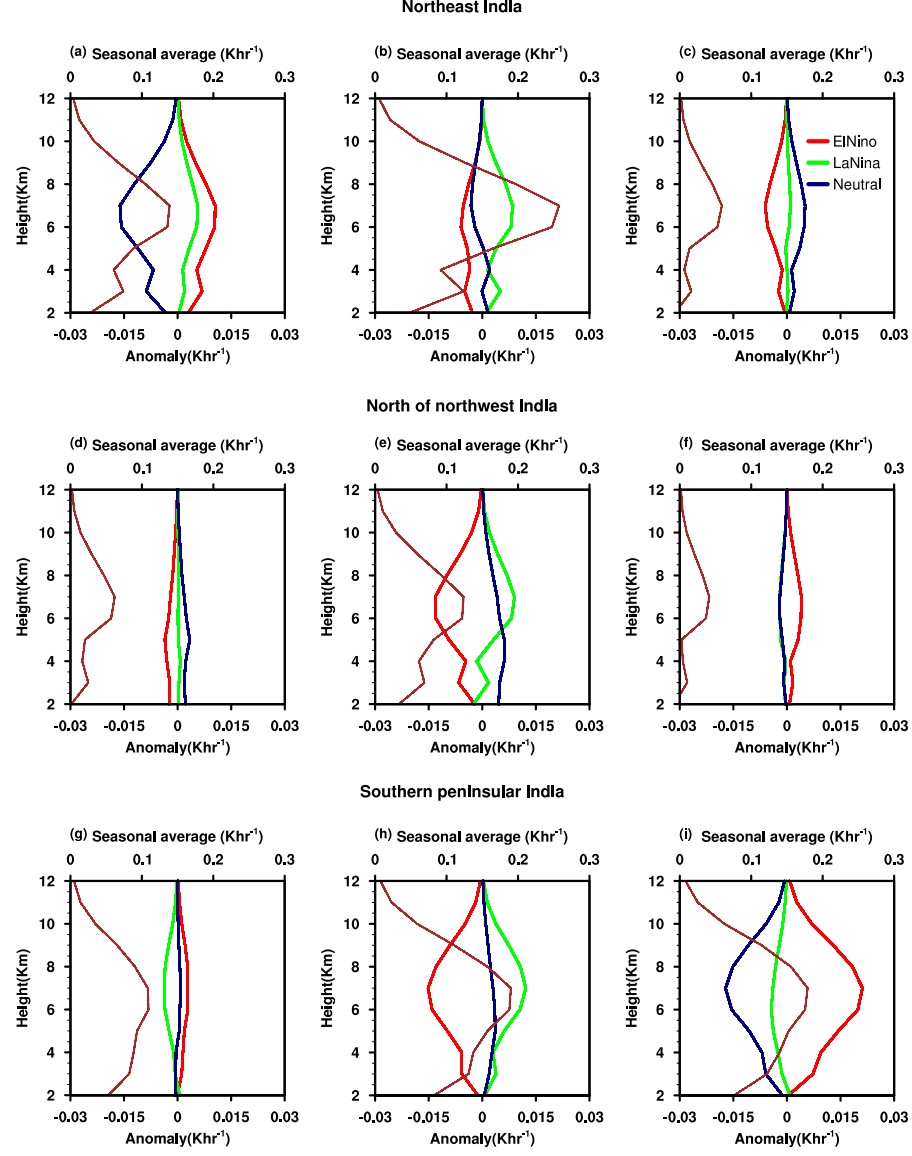

**Figure 7.** Seasonal average (brown dotted curve) and anomaly composite of latent heat during different ENSO phases. (a, d, g): pre-monsoon season, (b, e, h): Monsoon season, (c, f, i): post-monsoon season.

during the respective phases. By considering the anomalous circulation at 200 hPa level, an anomalous westerly (easterly) wind is prevalent over entire India during warm (cold) periods (Figure 9). Accordingly, upper-level wind pattern and variability of GPH together indicate the southward extension of WD during ENSO's warm phase. The sharp increase (decrease) of SH lies

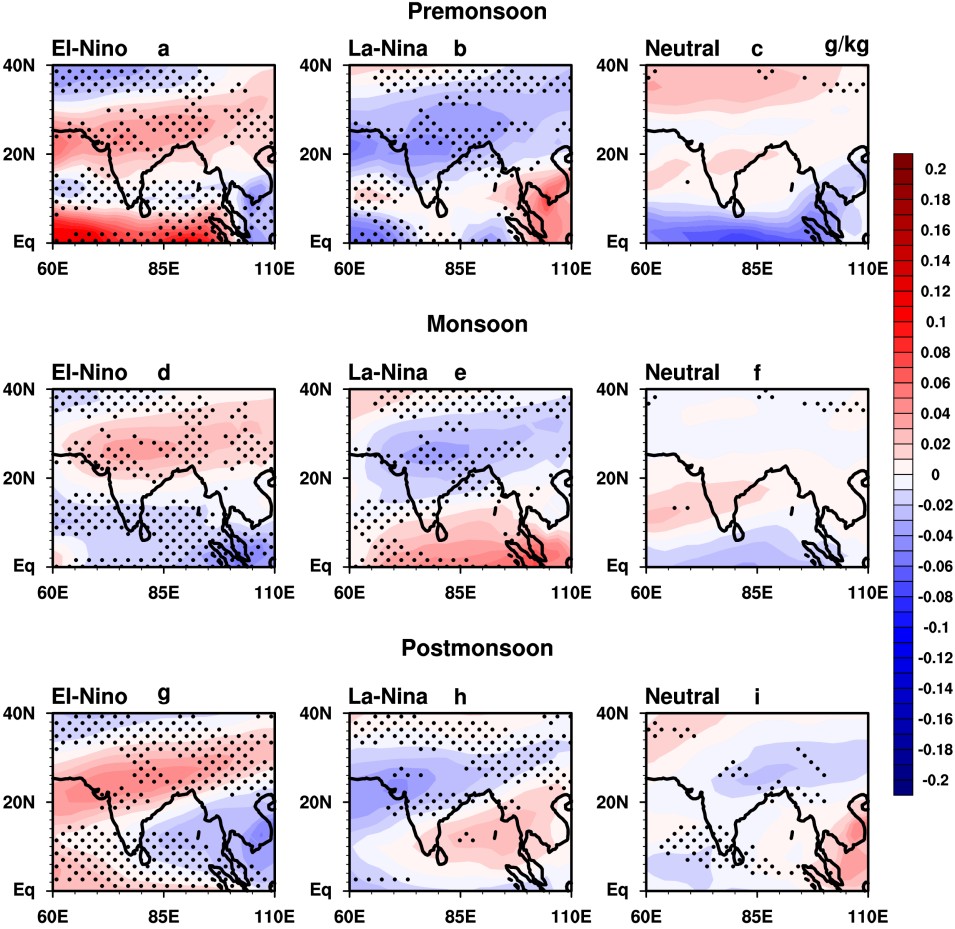

**Figure 8.** Anomaly composite of specific humidity at 300hPa during different ENSO phases with stippling to indicate statistically significant areas at a 95% confidence level.

precisely over the region of the maximum undulation of GPH over India during the warm (cold) phase (Figure 8 (g, h)). This suggests that ENSO indirectly influences the LFD over India during the post-monsoon season by modulating WDs' path.

## 4   Conclusion

In this study, we have discussed the influence of ENSO on LFD distribution during pre-monsoon, monsoon, and post-monsoon seasons over India. Regardless of ENSO phases, the LFD is peaking at the time of pre-monsoon season over NEI and SPI. However, the NNWI exhibits a peak LFD during the monsoon season. More importantly, the compelling correlation values

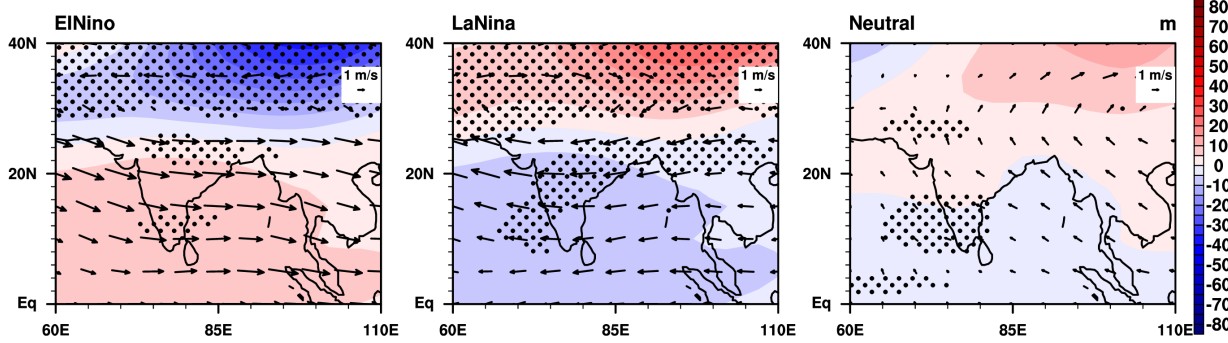

**Figure 9.** Anomaly composite of geopotential height at 500hPa and 200hPa wind during different ENSO phases. The stippling indicates statistically significant areas of geopotential height at a 95% confidence level.

indicates the solid linear dependence of LFD on graupel concentration over the hotspot regions of lightning. The LFD is in-
creased (decreased) compared to their average values over NEI and SPI during the warm (cold) phase of ENSO, and anomalies of the charge generating hydrometeors are also showing a similar kind of swing during the pre-monsoon season. An increase in graupel and snow formation above 6 km pinpoint that the warm phase of ENSO is conducive for deep convection over SPI during the pre-monsoon season. However, the neutral phase of ENSO favours the deepening of clouds over NNWI, as evidenced by the high values of the upper level-specific humidity.

During monsoon season, LFD over NEI and NNWI is higher than the average values during the La-Nina periods. The SPI is not showing significant variation in LFD with respect to different ENSO phases during the monsoon season. While considering the recent 12 years of this study, irrespective of the ENSO phases, every year has displayed above-average values of LFD over the NEI and NNWI region. Out of 9 years from 2005 to 2013, 8 displayed above-normal LFD over SPI, which signifies the intensification of LFD over the three hotspots during the monsoon seasons of the last decade.

Almost all regions in India are exhibiting higher LFD during the warm ENSO phase in the post-monsoon season. The elevated (reduced) GPH is visible all over India during the warm (cold) phase of ENSO, which is an indication of an increase (decrease) in the convective activity during the respective phases. Further, the intensification of convection during the warm phase is advocated by a significant rise of graupel and snow concentration over SPI. The entire years grouped under the warm phase of ENSO during the post-monsoon season show an increase of LFD over SPI, whereas the years elected under the
cold phase shows a disperse anomalous pattern. Both intensification and southward extension of WD is responsible for higher LFD over India in the warm phase, indicating indirect interaction between ENSO and LFD by modulating the mid-latitude westerlies.

*Data availability.* The LIS/OTD data and vertical profiles of hydrometeors, and latent heat obtained from the website http://ghrc.nsstc.nasa.gov/ and https://disc.gsfc.nasa.gov/datasets/ respectively. The GPH, wind and SH data are avilable the weblink https://psl.noaa.gov/data/gridded/ data.ncep.reanalysis.html. The HadISST data used in this work is accessible from the weblink https://psl.noaa.gov/gcos$_w$gsp/.

*Author contributions.* The paper and its methodology were conceptualized and developed by SA, AVS, and PV; AVS performed the analyses, and PV curated the data. The original draft preparation was by AVS; further reviewing and editing was by PV and SA. AVS handled visualization.

*Competing interests.* The authors declare that they have no known competing financial interests or personal relationships that could have appeared to influence the work reported in this paper.

*Acknowledgements.* We are grateful to NASA and PSL for providing LIS/ODT, TRMM, and NCEP Reanalysis data products, respectively, which are used in this study. Sreenath A V acknowledges Kerala State Council for Science, Technology, and Environment (KSCSTE), India, for providing financial support. Support from the Department of Atmospheric Sciences, Cochin University of Science and Technology is acknowledged.

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
