# Peer review of "Variability of lightning hazard over Indian region with respect to ENSO Phases"

_Natural Hazards and Earth System Sciences, 2020_

## Referee Comment (RC1) · Anonymous Referee #1 · 6 Dec 2020

The manuscript titled "Variability of lightning hazard over Indian region with respect to ENSO Phases" emphasizes the dependence of lightning activity over India with ENSO. Since lightning is a destructive natural hazard and India such a densely populated country, it is essential to understand the distribution of lightning flash rate (LFR) and its seasonal oscillation to avoid casualties. The manuscript is started by explaining the seasonal variability of LFR over India. Based on the seasonal change, three hotspots of LFR are identified. The variability of LFR under ENSO phases are substantiated with the vertical distribution of graupel and snow, responsible for the charge generation inside clouds. It is an excellent method to explain the anomalous distribution of LFR. The authors used the relatively less explored vertical profile of latent heating and hydrometeors TRMM data and presented deepening clouds based on this vertical profile.

Manuscripts display the distribution of LFR from 1995-2013 over the three hotspots and indicate that the three hotspots of LFR over the Indian land region became more prominent in the last decade of the monsoon season, one of the significant findings in this study, and it is useful for the society also. The indirect influence of ENSO warm phase on LFR over India during the post-monsoon season by modulating the subtropical westerly jet is another impressive result explained with geopotential height and upper-level wind anomaly.

Specific comments:

1. Include how the authors calculated the anomaly of LFR in the data and methodology section.

2. The location of monsoon trough and Western disturbances should be presented in Figure 2 for convenient reading.

Technical corrections:

There are some minor grammar issues in the manuscript. Eg.

Line 119: Change "enhance" to enhances

Line 137: "An elongated region, over central India,", remove the commas.

---

## Short Comment (SC1) · 13 Dec 2020

Sreenath A V

sreenathatmos@gmail.com

Specific comment 1: Include how the authors calculated the anomaly of LFR in the data and methodology section.

Response to comment 1: The term LFR anomaly indicates the difference in the composite of LFR during a particular ENSO phase in a specific season and the composite of LFR during all the three ENSO phases in that particular season. e.g., LFR anomaly during premonsoon during LaNina = (Composite of LFR during LaNina in premonsoon) - (Composite of LFR during all the three ENSO phases in premonsoon). The anomaly of all other parameters used in this study is calculated using the same method.

Specific comment 2: The location of monsoon trough and Western disturbances should

be presented in Figure 2 for convenient reading.

Response to comment 2: Corrections are included in figure 2.

Technical corrections: Line 119: Change "enhance" to enhances

Corrections are included: The entire years under the cold (warm) phase during pre-monsoon are showing a decrease (increase) of LFR over NEI (SPI) (Figure 3 (a, g)), which firmly indicates that the cold phase suppresses the LFR over NEI, and the warm phase enhances it over SPI.

Line 137: "An elongated region, over central India,", remove the commas.

Corrections are included: An elongated region over central India is showing higher (lower) LFR during the warm (cold) phase of ENSO (Figure 2 (d, e)).
* * *
[Figure]

**Fig. 1.** Anomaly composite of LFR during different ENSO phases. The box in figure (d) and (g) indicates the region of monsoon trough and western disturbance, respectively.

---

## Referee Comment (RC2) · Anonymous Referee #2 · 21 Jan 2021

Review of the paper: "Variability of lightning hazard over Indian region with respect to ENSO Phases", by Sreenath A V, Abhilash S, and Vijaykumar P The paper concerns the variation of the lightning flash activity over India and around during cycles of ENSO and at different seasons, pre-monsoon, monsoon and post-monsoon. Before to can evaluate the study and all comments in the paper, some information and clarifications are necessary. It is especially not clear how is calculated the anomaly for different parameters as LFR, graupel concentration at different seasons and in different regions. More specific comments are in the following. Major comments 1- The authors have to define the periods distinguished as premonsoon, monsoon and postmonsoon, because it is not made and it is a main information to well understand and interpret the values of the LFR provided in flash per day and per km2. What is the duration and the dates

of each season? At line 64 the data beginning is given in July 1995 and in Table 1, a value is given for each season of 1995. Does it mean the premonsoon is in July? 2- The LFR is calculated from the flashes detected by LIS or OTD. Is it estimated for the whole periods by extrapolation of the flashes detected by the sensors? Indeed, the sensors were above the region for a short time according to the low orbit satellite location. 3- At line 100, can we talk about three hot spots during the premonsoon season? In southern India it is not really a hot spot and in NNWI no more. "Hot spot" needs to be very distinctly higher than around. 4- Figure 2 needs explanation, how is calculated the anomaly? Why it is so different at NEI between El-Nino and La-Nina? While it is not very different for LFR in Figure 1 and even it seems LFR is larger in NEI during La-Nina while the anomaly is negative there. By comparing Figures 1a and 1b, I do not understand the differences between Figures 2a and 2b? I do not understand the negative value of anomaly in Figure 2b for NEI? (It is even lower than in Figure 2C for the same region NEI. Anyway, the values of the anomaly are < 0.015 for LFR values of about 0.1.. Is it significant? At line 115, it is said "the cold ENSO phase suppresses LFR over NEI" but in Figure 1b the hot spot over NEI is not suppressed at all! The other comment at line 115 can be discussed on the same way. Clarification is necessary, especially to describe the anomaly estimation. Same at line 119, it is impossible to say "which firmly indicates that the cold phase suppresses the LFR over NEI," according to Figure 1b. Same for comment about "the warm phase enhances it over SPI". 5- The same comments for the season "Monsoon" about the anomaly not obvious by looking at Figure 1d-e. They talk about increase of LFR along the coast of NWI at line 132 for El-Nino, not obvious in Figure 1d. At line 139: "The NEI is showing positive anomaly of LFR during both warm and cold phases of ENSO" this comment does not seem justified in Figure 2e, no increase. 6- The profiles in Figures 4-5-6 show concentration anomalies for graupel, snow and latent heat, respectively. The values seem very small to explain something. Why to not display the concentration directly? The values could be used to explain the storm occurrence? 7- Line 141: "The similarity in the LFR anomaly is noticeable in the distribution of graupel and snow

during the two phases (Figure 4 (b), 5 (b))." The anomaly is close to zero in this case for graupel and snow. Does it can explain the positive anomaly for the LFR during the two ENSO phases commented at line 139? Line 142 : Where is it visible that the LFR is suppressed in NWI for the warm season during monsoon? In Figure 1d it is well visible there is a hot spot like in cold season (Figure 1e).

Minor comments - The parameter LFR could be LFD as lightning flash density since it is a density (km-2 day-1). It is a daily density. Is it more consistent to talk about density? - Figure 1: the unit for the scale could more visible and written along the colored scale. - Line 72 : "seasons" - Line 90: With such average values of CAPE (1500 J/Kg) all over India, we can think storms are produced everywhere over India during that season. However, Figure 1 shows the LFR is < 0.05 km-2 day-1 over a large part of India. Do they want to say the CAPE is 1500 in average during the whole season or just during a short period? - Line 100: They could keep the same order for longitude/latitude in the definition of the regions. - Line 103: Is it better to use "neutral" and not "normal"? Check for others. - Line 105: I do not see a decrease of the LFR for El-Nino during the season pre-monsoon in NNWI (Figure 2a) while I see a decrease of LFR (negative anomaly) during La-Nina (Figure 2b). Can you check? - Line 110: Is LH defined before? (Latent Heat I suppose?). - Line 110: "decreases" since it is the amount? - Line 111: "unable" or "unbaled"? - Figure 3: Check the symbol of the variable on the graphs? It has to be LFR? - Figure 4 caption: graupel and not groupel. Figure 4h "India" - Figure 4 and Figure 5: Unit is g m-3 and not gm-3 but the values seem very low.. For graupel the maximum should be 0.0008 g m-3 i.e. 0.8 mg m-3 isn't that a little weak? -

---

## Short Comment (SC2) · 30 Jan 2021

Specific comment 1: How is calculated the anomaly for different parameters as LFR, graupel concentration at different seasons and in different regions.

Response to comment 1: The term Lightning Flash Rate (LFR) anomaly indicates the difference between the composite of LFR during a particular ENSO phase in a specific season and the composite of LFR during all the three ENSO phases in that particular season. e.g., LFR anomaly during pre-monsoon during La-Nina = (Composite of LFR during La-Nina in pre-monsoon) - (Composite of LFR during all the three ENSO phases in pre-monsoon). The anomalies of all other parameters used in this study are calculated using the same method. Thank you for the comment.

[Figure]

Major comment 1: The authors have to define the periods distinguished as premonsoon, monsoon and postmonsoon, because it is not made and it is a main information to well understand and interpret the values of the LFR provided in flash per day and per km2. What is the duration and the dates of each season? At line 64 the data beginning is given in July 1995 and in Table 1, a value is given for each season of 1995. Does it mean the premonsoon is in July?

Response to major comment 1: The pre-monsoon season includes the months of March, April, and May. The period June to September is the monsoon season. October, November and December months are taken as the post-monsoon season. The LFR data is available starting from July 1995 only. So the pre-monsoon season in our work starts from 1996 (March-April-May) and ends in 2013 (March-April-May). Due to data unavailability, the first monsoon season includes only three months (July, August, and September 1995). This particular season terminates in 2013 (June, July, August, and September). On the other hand, the post-monsoon season is prepared from 1995 (October, November, and December) to 2013 (October, November, and December). Since our work is based on the seasonal variation of LFR with ENSO Phases, the monthly average value of LFR from the LIS/OTD data product is used. The monthly average values outset from July 1995, and it concluded in December 2013. Perhaps our description in lines 64-65 may be a little confusing. The inclusion of the above information in the data and methodology section will provide better clarity for readers. Thank you for this suggestion. Table 1 provides the ONI index to show the pre-monsoon season in 1995 was the normal phase of ENSO, and it doesn't mean July is in pre-monsoon, and it doesn't mean the pre-monsoon started in 1995. In figure 3(a), (d), and (g), you can see that LFR values are absent during the 1995 pre- monsoon season.

Major comment 2: The LFR is calculated from the flashes detected by LIS or OTD. Is it estimated for the whole period by extrapolation of the flashes detected by the sensors? Indeed, the sensors were above the region for a short time according to the low orbit

satellite location.

Response to major comment2: Combined OTD (Optical Transient Detector) + LIS (Lightning Imaging Sensor) monthly averaged flash rates expressed as flash rate density (flash km-2day -1 ) available from http://ghrc.nsstc.nasa.gov/ is used in this work. These products compute mean lightning flash rates by accumulating the total number of flashes observed and the total observation duration for each grid box (2.5° × 2.5°) from the thousands of individual satellite orbits. The lightning climatology derived from OTD / LIS (Cecil et al., 2014) provides a unique observational basis for the global flash distribution in monthly time series (Kamra and Athira., 2016), seasonal cycles (Christian et al., 2003), or diurnal cycles (Blakeslee et al., 2014). To produce the low-resolution monthly time series (LMRTS) data, LIS and OTD flash rates and view times are smoothed precisely and are extracted for the middle day of each month (Cecil et al., 2014). The lightning flash rates in an LRMTS have slightly over three months of temporal smoothing and 7.5°*7.5° spatial smoothing (Cecil et al., 2014). The data sets are described in greater detail in the following paper: Gridded lightning climatology from TRMM-LIS and OTD: Dataset description by Cecil et al. (2014).

Major comment 3: At line 100, can we talk about three hot spots during the premonsoon season? In southern India it is not really a hot spot and in NNWI no more. "Hot spot" needs to be very distinctly higher than around.

Response to major comment 3: Ahmad and Ghosh (2017) reported that compared to other regions of India, lightning activity is higher over the North-Eastern part and southern part of India during pre-monsoon season in India. They also observed that the maxima of lightning during post-monsoon is also lying over the southern and eastern regions of India. So previous studies indicate that we can take the southern peninsula as one of the hotspots of lightning over India. As you pointed out, the hotspot is not distinguishable in figure 1, probably due to the color scale we have chosen. Thank you very much for this useful remark. Accordingly, we have modified Figure 1 with an appropriate color scale to get a better view of the region corresponding to a higher

lighting occurrence (Please see the revised figure 1 given below). NNWI is the acronym for north of northwest India. You can also see a dominance of lightning over the same region during pre-monsoon, monsoon, and post-monsoon seasons in India.

Major comment 4: Figure 2 needs explanation, how is calculated the anomaly? Why it is so different at NEI between El-Nino and La-Nina? While it is not very different for LFR in Figure 1 and even it seems LFR is larger in NEI during La-Nina while the anomaly is negative there. By comparing Figures 1a and 1b, I do not understand the differences between Figures 2a and 2b? I do not understand the negative value of anomaly in Figure 2b for NEI? (It is even lower than in Figure 2C for the same region NEI. Anyway, the values of the anomaly are < 0.015 for LFR values of about 0.1.. Is it significant? At line 115, it is said "the cold ENSO phase suppresses LFR over NEI" but in Figure 1b the hot spot over NEI is not suppressed at all! The other comment at line 115 can be discussed on the same way. Clarification is necessary, especially to describe the anomaly estimation. Same at line 119, it is impossible to say "which firmly indicates that the cold phase suppresses the LFR over NEI," according to Figure 1b. Same for comment about "the warm phase enhances it over SPI".

Response to major comment 4: As explained in response to the specific comment 1, the LFR anomaly is calculated by taking the difference between the composite of LFR during a particular ENSO phase in a specific season and the composite of LFR during all the three ENSO phases in that particular season. e.g., LFR anomaly during pre-monsoon during LaNina = (Composite of LFR during LaNina in pre-monsoon) - (Composite of LFR during all the three ENSO phases in pre-monsoon). The disparity of LFR over north east India (NEI) during El-Nino and La-Nina may arise due to the difference in convective clouds' intensity over that region. The hydrometeors (Snow and Graupel) and the latent heat profile support the higher LFR during the El-Nino phase during the pre-monsoon season (Figure 4 (a), 5(a), and 6(a)). In contrast, the hydrometeors concentration and latent heat release are less during the La-Nina period, which indicates suppression of convective clouds and hence the LFR. The contrast

between figure1 (climatology of LFR) and figure 2 (Anomaly of LFR) must have evolved due to a plotting error that happened unknowingly while drawing figure 1. We are very much thankful to the reviewer for pointing this out. We have modified Figure 1 and given below. In our study, the LFR varies from 0.02 to 0.2 flash/km*2/day (Figure 1). A prior study by Kmara and Athira (2016) shows that LFR ranges between 0.01 to 0.2 flash/km*2/day over India. The main regions of lightning over India (e.g., NEI) shows an increase of more than 0.01 flash/km*2/day during El-Nino and a decrease of more than 0.01 flash/km*2/day during La Nina in pre-monsoon season (Figure 2). This indicates that the modulus of the anomaly over these hotspots regions is higher than the LFR climatology over many parts of India. Yuan et al. (2016) calculated the LFR anomaly (Unit: Flashes/Km*2/month) over Southeast Asia during El-Nino and La-Nina episodes. They found a similar range of LFR anomalies as in our study. Further, their analysis is based on this anomalous pattern of LFR. Hence, previous studies support that the anomalous values of LFR as shown in our study are sufficient to explain the contrast in LFR over northeast India, north of northwest India, and southern peninsular India during different ENSO phases. An examination of figure 1, along with figure 2 (a, b), is helpful. In figure 2 (a, b), one can notice that NEI shows reduced LFR during the La-Nina phase (cold phase) of ENSO, and the El-Nino phase (Warm phase) shows enhanced LFR over southern peninsular India.

Major comment 5: The same comments for the season "Monsoon" about the anomaly not obvious by looking at Figure 1d-e. They talk about increase of LFR along the coast of NWI at line 132 for El-Nino, not obvious in Figure 1d. At line 139: "The NEI is showing positive anomaly of LFR during both warm and cold phases of ENSO" this comment does not seem justified in Figure 2e, no increase.

Response to major comment 5: The anomaly figure suggests that (Figure 2(a)) LFR shows an increase (although small in magnitude (between 0.002 - 0.006)) along the northwest coast of India. The anomaly plot better serves this purpose of capturing small changes in LFR with different ENSO phases. At the same time, climatology

figure 1 is used to get an overall idea about the distribution of LFR. Small variations in LFR values are not discernible in the climatology figure 1. Anomaly plots are widely used as a tool to overcome this difficulty. In the present work, NEI is defined as the region between 20° - 30° N and 85°-95° E. Kindly find the anomaly plot in which NEI is marked inside a black box (Figure 2(e)), which shows that figure 2(e) can justify the comment we made in line 139: i.e., "The NEI is showing positive anomaly of LFR during both warm and cold phases of ENSO."

Major comment 6: The profiles in Figures 4-5-6 show concentration anomalies for graupel, snow and latent heat, respectively. The values seem very small to explain something. Why to not display the concentration directly? The values could be used to explain the storm occurrence?

Response to major comment 6: Graupel and snow are the different forms of ice content inside the convective clouds. In situ airborne observations during the Cloud-Aerosol Interaction and Precipitation Enhancement Experiment (CAIPEEX) over various locations of India shows that convective clouds during the pre-monsoon and monsoon period have an ice water content of 10-4 to 1 g m-3 (Patade et al., 2015). Moreover, in situ measured ice cloud properties in the European Cloud Radiation Experiment (EU-CREX) have reported a similar range of ice water content inside the clouds system (10-4 g m-3 to 1 g m-3) (Hogan et al., 2006). So the anomaly values of snow or graupel are significant compared to actual values of ice content inside the cloud system. As said earlier, anomaly values are used to catch the minute variations in the vertical distribution of hydrometeors during different ENSO phases, which will help better understand the variability in the hydrometeors profiles. Kumar et al. (2013) analyzed the contrast between cloud properties over India's west coast and the Myanmar coast. They provide the vertical profile of latent heating over the two contrasting regions expressed in degrees/day, ranging between 0 to 0.3. The west coast of India is marked by shallow convection, while vigorous convective clouds dominate Myanmar coasts during the Asian summer monsoon. In the present work, anomalous latent heating exists mostly

between +/- 0.01, and in some cases, it is extending up to +/-0.02 (Kelvin/hr). Hence, from prior studies, the anomaly values of latent heating are highly significant compared to the actual values of the same, and they can indicate modulations of convective cloud formation and storm occurrence along with hydrometeors profiles in different seasons of India with various phases of ENSO.

Major comment 7: Line 141:"The similarity in the LFR anomaly is noticeable in the distribution of graupel and snow during the two phases (Figure 4 (b), 5 (b))." The anomaly is close to zero in this case for graupel and snow. Does it can explain the positive anomaly for the LFR during the two ENSO phases commented at line 139? Line 142 : Where is it visible that the LFR is suppressed in NWI for the warm season during monsoon? In Figure 1d it is well visible there is a hot spot like in cold season (Figure 1e).

Response to major comment 7: During monsoon season, the anomalous snow and graupel distribution unable to provide strong evidence for the increase of LFR during the warm and cold phases of ENSO over NEI. We have used figure 3 to ensure the above observation by indicating that all the years under the warm phase and the majority of years under the cold phase (during the monsoon season) show an increase in LFR over NEI. In our study, the region between 25°N - 40° N and 65°E - 80°E is defined as north of northwest India (NNWI). From figure 2(d) (given inside a red box), you can see the suppression of LFR over this region in the warm phase during the monsoon season. As same as in figure 1 (d), you can see a hotspot like region of LFR over the same area in figure 1 (e) and figure 1(f). As mentioned earlier, these figures represent the climatology pattern of LFR. Its purpose is to provide an overall idea about the distribution of LFR in different seasons with various ENSO phases. To get the idea about the modulation of LFR with different ENSO phases during different seasons, we can use the anomaly plot.

Minor comment 1: The parameter LFR could be LFD as lightning flash density since it is a density (km-2 day-1). It is a daily density. Is it more consistent to talk about

density?

Response to minor comment 1:We have defined lightning flash rate as the number of flashes/km2/day in this work. Some earlier studies also expressed this parameter as the flash rate (Kamra and Athira (2016); Yuan et al., (2016)). Since the previous works define the term as the flash rate, we used the same term. Further, the term 'rate' is more appropriate as it appears to be more synonymous with 'the chances of occurrences for lightning flashes' rather than to the intensity of individual flashes. On the other hand, the term 'density' may imply the intensity or severity of lightning, which we do not intend to suggest in the present study.

Minor comment 2: Figure 1: The unit for the scale could more visible and written along the colored scale.

Response to minor comment 2: Correction included. Please see figure 1.

Minor comment 3: Line 72 : "seasons" -

Response to minor comment 3: Correction included.

Minor comment 4: Line 90: With such average values of CAPE (1500 J/Kg) all over India, we can think storms are produced everywhere over India during that season. However, Figure 1 shows the LFR is < 0.05 km-2 day-1 over a large part of India. Do they want to say the CAPE is 1500 in average during the whole season or just during a short period ?

Response to minor comment 4: A typing error occurred in line 90. Thanks for pointing out. Now the sentence is rewritten into "The seasonal average of convective available potential energy (CAPE) highest over the east coast of India, and the same is 1500 J/kg all over south India (Murugavel et al., 2014). At the same time, large regions of India, especially the central Indian region, show a seasonal average of CAPE less than 1000 J/Kg (Murugavel et al., 2014)." By including this correction, we can explain the existence of LFR < 0.05 km-2 day-1 over a large part of India.

Minor comment 5: Line 100: They could keep the same order for longitude/latitude in the definition of the regions.

Response to minor comment 5: To get more accurate results in our work, we have chosen the latitude and longitude for the regions around the maximum values LFR (we called it hotspots of LFR). Figure 1(a) shows that a hotspot of LFR over NEI lies exactly inside 20° N-30° N and 85° E- 195° E. NNWI is exhibiting maximum values of LFR between 25° N-40° N and 65° E-80° E (Figure 1(d) ). The intense LFR activity over southern peninsular India (SPI) includes the region between 5° N-15° N and 75° E-80° E (figure 1(a)). Other areas around SPI have smaller values of LFR as similar to the majority of regions in India. We cannot consider those regions as a hotspot. The same is valid in the case of NEI and NNWI.

Minor comment 5: Line 103: Is it better to use "neutral" and not "normal"? Check for others.

Response to minor comment 5: Correction included. Thank you for this suggestion.

Minor comment 6: Line 105: I do not see a decrease of the LFR for El-Nino during the season premonsoon in NNWI (Figure 2a) while I see a decrease of LFR (negative anomaly) during La-Nina (Figure 2b). Can you check?

Response to minor comment 6: Please see figure 2(a), in which a decrease of LFR can be seen over NNWI, which is enclosed inside a green box, and this result is supported by the anomalous decrease of graupel conecntration during premonsoon season in El-Nino phase (Figure 4(d)).

Minor comment 7: Line 110: Is LH defined before? (Latent Heat I suppose?).

Response to minor comment 7: LH denotes latent heat. Correction included. Thank You.

Minor comment 8: Line 110: "decreases" since it is the amount?

Response to minor comment 8: Correction included. Thank You.

Minor comment 9: Line 111: "unable" or "unbaled"?

Response to minor comment 9: Correction included. Thank You.

Minor comment 10: Figure 3: Check the symbol of the variable on the graphs? It has to be LFR?

Response to minor comment 10: Correction included. The symbol is changed to LFR. Please see figure 3. Thank You.

Minor comment 11: Figure 4 caption: graupel and not groupel.

Response to minor comment 11: Correction included. Thank You.

Minor comment 12: Figure 4h "India"

Response to minor comment 12: Correction included. Please see figure 4(h). Thank You.

Minor comment 13: Figure 4 and Figure 5: Unit is g m-3 and not gm-3 but the values seem very low.. For graupel the maximum should be 0.0008 g m-3 i.e. 0.8 mg m-3 isn't that a little weak?

Response to minor comment 13: Cloud ice water content (IWC) is defined as cloud ice mass in the unit volume of atmospheric air. As we indicated in the response of major comment 6, convective clouds during the pre-monsoon and monsoon periods have an ice water content (graupel and snow are the ice forms inside the clouds) of 10-4 to 1 g m-3 (Patade et al., 2015). ie, 0.1 mg-1000 mg. So, the anomaly values are significant in comparison with actual values of ice content inside the clouds.

Reference

Ahmad, A., & Ghosh, M. (2017). Variability of lightning activity over India on ENSO time scales. Advances in Space Research, 60(11), 2379-2388.

Blakeslee, R. J., Mach, D. M., Bateman, M. G., & Bailey, J. C. (2014). Seasonal variations in the lightning diurnal cycle and implications for the global electric circuit. Atmospheric research, 135, 228-243.

Cecil, D. J., Buechler, D. E., & Blakeslee, R. J. (2014). Gridded lightning climatology from TRMM-LIS and OTD: Dataset description. Atmospheric Research, 135, 404-414.

Christian, H. J., & Blakeslee, R. J. D. j. Boccippio, WL Boeck, DE Buechler, KT Driscoll, SJ Goodman, JM Hall, WJ Koshak, DA Mach and MF & Stewart (2003):"Global frequency and distribution of lightning as observed from space by the Optical Transient Detector". J. Geophys. Res, 108, 4005.

Hogan, R. J. MP Mittermaier, and AJ Illingworth, 2006: The retrieval of ice water content from radar reflectivity factor and temperature and its use in evaluating a mesoscale model. J. Appl. Meteor. Climatol, 45, 301-317.

Kamra, A. K., & Athira, U. N. (2016). Evolution of the impacts of the 2009–10 El Niño and the 2010–11 La Niña on flash rate in wet and dry environments in the Himalayan range. Atmospheric Research, 182, 189-199.

Kumar, S., Hazra, A., & Goswami, B. N. (2014). Role of interaction between dynamics, thermodynamics and cloud microphysics on summer monsoon precipitating clouds over the Myanmar Coast and the Western Ghats. Climate dynamics, 43(3-4), 911-924.

Patade, S., Prabha, T. V., Axisa, D., Gayatri, K., & Heymsfield, A. (2015). Particle size distribution properties in mixed‐phase monsoon clouds from in situ measurements during CAIPEEX. Journal of Geophysical Research: Atmospheres, 120(19), 10-418.

Yuan, T., Di, Y., & Qie, K. (2016). Variability of lightning flash and thunderstorm over East/Southeast Asia on the ENSO time scales. Atmospheric Research, 169, 377-390.
* * *
**NHESSD**

[Figure]

[Figure]

**Fig. 1.** LFR climatology during different ENSO phases

[Figure]

**Premonsoon**

(a) El-Nino

(b) La-Nina

(c) Neutral    count/km^2/day

**Monsoon**

(d) El-Nino

(e) La-Nina

(f) Neutral

**Postmonsoon**

(g) El-Nino

(h) La-Nina

(i) Neutral

**Fig. 2.** Anomaly composite of LFR during different ENSO phases.

[Figure]

**Fig. 3.** The anomaly of LFR during the individual years with different ENSO phases. Red color label bar corresponds to warm (El-Nino); green one corresponds to the cold (La-Nina) and blue color label bar indi

**North-East India**

(a) Pre-monsoon

(b) Monsoon

(c) Post-monsoon

ElNino
LaNina
Normal

(d)

(e) **North-West India**

(f)

(g)

(h) **South India**

(i)

**Fig. 4.** Anomaly composite of graupel during different ENSO phases

---

## Author Comment (AC1) · 8 Feb 2021

Specific comment 1: Include how the authors calculated the anomaly of LFR in the data and methodology section.

Response to comment 1: The term LFR anomaly indicates the difference in the composite of LFR during a particular ENSO phase in a specific season and the composite of LFR during all the three ENSO phases in that particular season. e.g., LFR anomaly during premonsoon during LaNina = (Composite of LFR during LaNina in premonsoon) - (Composite of LFR during all the three ENSO phases in premonsoon). The anomaly of all other parameters used in this study is calculated using the same method.

Specific comment 2: The location of monsoon trough and Western disturbances should

be presented in Figure 2 for convenient reading.

Response to comment 2: Correction included. The monsoon trough and western disturbances are represented inside a black box in figure 2(d) and figure 2(d), respectively. Please see the revised figure. Thank you.

Technical corrections:There are some minor grammar issues in the manuscript. Eg.Line 119: Change "enhance" to enhances Line 137: "An elongated region, over central India,", remove the commas.

Response to technical corrections: Correction included. Thank you.
* * *
[Figure]

**Premonsoon**

**Monsoon**

**Postmonsoon**

**Fig. 1.** Anomaly composite of LFR during different ENSO phases. The box in figure (d) and (g) indicates the region of monsoon trough and western disturbance, respectively.

---

## Author Response (AR2)

**Responses to the referee comments**

We thank the reviewer for his/her insightful comments and suggestions. We have modified the figures and text as directed by the reviewer. Below is our point-wise response to the specific comments raised in the second review.

**Major comment 1:** One of the change concerns figure 1 for the LFR values. Apparently, there was an error in the plot, especially in the phase La-Nina. The LFR is now much weaker. Of course, it clarifies the comments, but is a little surprising to have made comments that were not consistent with the figure.

**Response to major comment 1**: LFR is weaker only during the post-monsoon season in India. Previous studies also reported a similar LFR distribution (Kamra et al., 2014). Those comments were made for anomalous lightning flash rate (LFR) in figure 2, and concerning figure 2, the description was reasonable in terms of anomaly. Figures 1 and 2 were inconsistent during the initial submission stage because of the error in figure 1. Accordingly, we rectified those mistakes during the second revision and believe that the revised description is robust for the figures. We are immensely thankful for these comments, which have helped to improve our manuscript significantly.

**Major comment 2:** Another change is the colored scale with a degraded resolution. It is not easy to appreciate some low values difference although the authors made comments about low values as for example in the case of post-monsoon period. The Indian peninsula at that moment is concerned by very weak lightning activity and it is difficult to see a difference between the different regions and the different ENSO phases.

**Response to major comment 2**: Correction included, and the colour scale is adjusted to reflect the prominent features of regions having maximum LFR. The edited version of the manuscript provides figure 1 with a colour scale with an enhanced resolution. The text is also changed according to your suggestions. Kindly see the revised manuscript.

**Major comment 3**: We can also wonder why the authors choose to plot such a large area for the maps of the LFR since they make comments and analysis about India? It is especially true for the south part of the maps and for the western part too. With a reduced area, the visibility should be better?

**Response to major comment 3**:  Correction included. As suggested by the reviewer, we have restricted the geographical area to 60-110º E, and 0-40º N for better visibility of the prominent features of LFR discussed in the text.

**Major comment 4:** The first comment in the abstract at line 10 about three hotspots: "three hotspots of LFR over the Indian land region became more prominent in the last decade of the monsoon season" is not obvious. I do not see three hotspots?

**Response to major comment 4:**

We modified this sentence to "It is striking to note that there are three hotspots of lightning flash density (LFD) over the Indian land region, which became more prominent during the monsoon seasons of last decade." The hot spots regions discussed here is in agreement with earlier studies. Ahmad and Ghosh (2017) reported that lightning activity is higher over the North-Eastern part and southern part of India during the pre-monsoon season than in other regions of India. They also observed that the maxima of lightning during post-monsoon is also lying over India's southern and eastern areas. Saha et al. (2017) confirm that the north-western and north-eastern regions of India and the southern tip of peninsular India are the three main zones prone to deep convection.

The three hotspots of LFD discussed in the manuscript is more discernible from modified figure 1 and highlighted as coloured boxes. The LFD becomes more prominent in the last decade of the monsoon season, as evident from the area-averaged anomaly presented in figure 4.

**Major comment 5:** Lines 150-155: about the values of ice particle concentration, the value range is wide but for convective clouds we can suppose large values within the range are more probable. The range is presented with this interval [10-4 – 1 g m-3] which corresponds with values measured during a campaign. My feeling is that most values are in the upper part of the interval. Thus, as I noted in the first review, the values for the anomalies in figures 4 and 5 are < 0.0005 g m-3. Again, I do not understand such low values when the concentration is close to 1 g m-3. What does it mean? Are these values valuable for the convective clouds? They are issued from the NCEP/NCAR database with a resolution of 0.5°x0.5° but at which time do they correspond? How are these values representative of the convective clouds when they occur? For the case of the NNWI region and pre-monsoon season, the LH anomaly is < 0.01 °/hr, it is also very low values.

**Response to major comment 5:** This study aims to understand the seasonal variability of LFD over the Indian region with respect to different ENSO phases. Hence this study utilized monthly mean values of LFD and cloud hydrometeors from TRMM observations. We agree that the values are insignificant as far as individual clouds are concerned but significant for seasonal composite analysis. The area averaging also reduces the absolute values as compared to individual cloud cases. Monthly averaged TRMM-3A12 data is available for graupel and snow from January 1998 to December 2013 with a spatial resolution of 0.5*0.5 degrees. The parameter averaged for the premonsoon (March-May), monsoon (June-sept) and post-monsoon (October-December) season from 1998 to 2013 with respect to La-Nina, El-Nino and Neutral phases of ENSO are used in this work. Anomalies of these parameters are calculated in the following way.

Graupel/snow anomaly in this study indicates the difference between the composite of graupel/snow concentration during a particular ENSO phase in a specific season and the composite of graupel/snow concentration during all three ENSO phases for that particular season. e.g., Graupel/snow anomaly during pre-monsoon during La-Nina = (Composite of graupel/snow concentration during La-Nina in pre-monsoon) - (Composite of graupel/snow concentration during all the three ENSO phases in pre-monsoon). The seasonal average and anomaly of latent heat with respect to ENSO phases are also calculated similarly.

According to CAIPEEX measurement, Patade et al. (2015) showed that ice particle concentration inside the convective system varies with seasons. During the pre-monsoon season, they found that most values exist between $10^{-2}$ and $10^{-3}$. While their values mainly exist between 10-1 to 10-3 during monsoon and post-monsoon season. So it is not easy to say that ice concentration is close to the upper part of the interval. From TRMM observations and high-resolution model simulations, Abhilash et al. (2008) reported ice concentrations of $10^{-2}$ to $10^{-3}$ for convective storms over the Indian region.

For signifying the importance of anomalous concentration, we have included the seasonal average of graupel and snow content with the ENSO phase in the modified figures. From these revised figures, we can see that actual values of the seasonal average of graupel with respect to the ENSO phase are less than 2 mg.m$^{-3}$ over NEI, NNWI and SPI. The same is true for the seasonal average of snow content and suggests that anomalous values are significant compared to their actual values.

LH anomaly is < 0.01 in magnitude, and their absolute values are also less when averaging over a large domain, and the sentence is now modified accordingly. Hence while averaging over the season and over a larger region, we can expect values less than one order of magnitude as compared to individual clouds averaged over a small region.

**Major comment 6:** Line 163, the authors write: "the cold ENSO phase suppresses LFR over NEI and SPI with enhanced LFR over CI (Figure 2 (b))". It is in contradiction with line 109, when they wrote "Irrespective of ENSO phases, the LFR peak is located over northeast India (NEI) during the pre-monsoon season:" They cannot say the LFR is suppressed during the pre-monsoon season and the cold ENSO phase, by looking at Figure 1, even if the anomaly is negative! The LFR is displayed in Figure 1. The LFR in NEI is between 0.08 and 0.12 according to Figure 1b for the cold phase and the anomaly according to Figure 2 is between -0.01 and -0.014, it is about 10% of the LFR value.

They cannot talk about "suppress" it is only lower than the average between the three ENSO phases if I understand well.

**Response to major comment 6:** Correction included. The sentence is rewritten according to your suggestion. Thank you for the valuable feedback.

**Major comment 7:** Lines 162-168: For the comparison of different ENSO phases the mirror image effect is not surprising since the anomaly is calculated by the difference with the average value. If one phase involves a decrease another (or both others) has to involve an increase according the definition of the anomaly. Again, at line 168, they cannot say "that the cold phase suppresses the LFR over NEI" since it is not suppressed. Anyway, if they talk about an increase or a decrease, they have to quantify it to discuss the relative variation.

**Response to major comment 7:** Correction included. The wordings are changed according to your suggestion. Thank you.

**Major comment 8:** Line 169: Figure 4a shows the anomaly of graupel concentration in NEI and during the pre-monsoon season. According to the definition of the anomaly as a difference to the mean value for each phase, the sum of the anomaly values must be zero. Apparently, it is not the case for this graupel concentration. The same comment can be made for other panels of Figure 2: d, e, h, l and the same for Figure 5 and Figure 6. Can the authors explain these results? Anyway, for the parameters of microphysics issued from re-analysis, the description added in the new version of the paper is insufficient to understand the signification of these concentrations. Are they average and for which time and location are they representative?

**Response to major comment 8:** Thank you for this valuable suggestion. We rechecked the data and found that some missing values in the data set created this problem. Now it is corrected and the figures are redrawn. Please see the revised figures 2, 5 and 6. We did a correlation analysis between lightning flash rates with microphysical parameter (graupel concentration) and included in the revised manuscript with a detailed explanation.

Since the data sets are only available from January 1998, the microphysical parameter is averaged for the pre-monsoon (March-May), monsoon (June-September) and post-monsoon season (October-December) from 1998 to 2013 with respect to La-Nina, El-Nino and Neutral phases of ENSO over NEI (85˚ E-95˚ E, 20˚ N-30˚ N), NNWI (25˚ N-40˚ N, 65˚ E-80˚ E) and SPI (5˚ N-15˚ N, 75˚ E-80 ˚ E).

The authors are obliged to the reviewer for pointing out this important issue and allowing us to incorporate the corrections, and now the figures maintain the budget.

**Major comment 9:** For section 3.3, it is a little the same problem with the comparison of region NNWI during the different phases. They cannot use "suppress" for the LFR and again, a quantitative analysis could be made.

**Response to major comment 9:** Correction included. Thanks for the suggestion.

**Minor comments:**

**Minor comment 1:** I said in my first review the parameter LFR could be LFD as lightning flash density since it is a density (km-2 day-1). It is a daily density. Is it more consistent to talk about density? As in Albrecht et al. (2016) a combination is used when the double scale (time and space) is used for the flash count: for example "The TRMM LIS total lightning flash rate density (FRD – fl km-2 yr-1)" is generally used at the scale of the year. Ref Albrecht, R., Goodman, S., Buechler, D., Blakeslee, R., and Christian, H.: Where are the lightning hotspots on Earth?, Bull. Amer. Meteor. Soc., 97, 2051-2068, doi:10.1175/bams-d-14-00193.1, 2016.

It is also the case in Christian et al. (2003): in the abstract: "The Congo basin, which stands out year-round, shows a peak mean annual flash density of 80 fl km-2 yr-1 in Rwanda, and includes an area of over 3 million km2 exhibiting flash densities greater than 30 fl km-2 yr-1 (the flash density of central Florida)." It sounds better with "density", but if the authors prefer "rate" it is also possible, some authors use it, as Cecil et al. for example.

**Response to minor comment 1:** Thank you for this suggestion. We admit that initially, we were a bit confused on which term is more accurate and now realize that 'flash density' is better usage than flash rate. We have changed LFR to LFD throughout the manuscript. Thank you very much.

**Minor comment 2:** Line 20-22: the annual death rate has to be 2,266 with a total of 31,725 in 14 years?

**Response to minor comment 2 :** Correction included. Thank you.

**Minor comment 3:** Line 23: "they find" and not "they finds"

**Response to minor comment 3:** Correction included. Thank you.

**Minor comment 4:** Line 77: the power numbers at exponent for the units of flash rates

**Response to minor comment 4:** Correction included. Thank you.

**Minor comment 5:** Line 83: LRMTS

**Response to  minor comment 5:** Correction included. Thank you.

**Minor comment 6:** Line 104: Write "If the ONI value is above (below) +0.5° (-0.5°) C…"
**Response to  minor comment 6:** Correction included. Thank you.

**Minor comment 7:** Line 137: Is it possible to talk about three hotspots? That in the southern part of Indian Peninsula is not very visible and the LFR does not reach high values there. It was not clear in the initial maps with a better color resolution (figure 1 of the previous version), it seems there is an effect of amplifying with the new color scale.

**Response to  minor comment 7:**  Ahmad and Ghosh (2017) reported that lightning activity is higher over the North-Eastern part and southern part of India during the pre-monsoon season than in other regions of India. They also observed that the maxima of lightning during post-monsoon is also lying over India's southern and eastern areas. Similarly, we are getting higher LFR over NEI, NNWI and SPI. We have enhanced the resolution of the colour scale of figure 1 in this revised version and now higher LFD is better visible over SPI. Please see the revised figure 1 in the manuscript. Thank you.

**Minor comment 8:** Line 141: The comment is about the values of LFR that are low during the pre-monsoon, therefore the figure for reference is Figure 1 that displays directly the LFR values. Figure 2 displays the anomaly that is a comparison with other phases of ENSO. Even if the anomaly is negative (positive) it does not mean the LFR is low (large).
**Response to  minor comment 8:** Correction included. Thank you.

**Minor comment 9:** Line 142: Since the anomaly is the comparison between the different ENSO phases for a given season, a negative anomaly in a region for one phase implies a positive anomaly for another phase. Therefore, why to say "however" at the beginning of the sentence since it is an evidence?
**Response to  minor comment 9:** Correction included, text modified. Thank you.

**Minor comment 10:**  Lines 142-144: the sentence needs to be referred to the figure 3 and to say at which region it is applied.
**Response to  minor comment 10:** Correction included. Thank you.

**Minor comment 11:** Line 189: "analyzing"

**Response to minor comment 11:** Correction included. Thank you.

**Reference**

Abhilash, S., Mohankumar, K., & Das, S. (2008). Simulation of microphysical structure associated with tropical cloud clusters using mesoscale model and comparison with TRMM observations. International Journal of Remote Sensing, 29(8), 2411-2432.

Ahmad, A., & Ghosh, M. (2017). Variability of lightning activity over India on ENSO time scales. Advances in Space Research, 60(11), 2379-2388.

Kamra, A. K., & Athira, U. N. (2016). Evolution of the impacts of the 2009–10 El Niño and the 2010–11 La Niña on flash rate in wet and dry environments in the Himalayan range. Atmospheric Research, 182, 189-199.

Patade, S., Prabha, T. V., Axisa, D., Gayatri, K., & Heymsfield, A. (2015). Particle size distribution properties in mixed-phase monsoon clouds from in situ measurements during CAIPEEX. Journal of Geophysical Research: Atmospheres, 120(19), 10-418.

Saha, U., Siingh, D., Midya, S. K., Singh, R. P., Singh, A. K., & Kumar, S. (2017). Spatio-temporal variability of lightning and convective activity over South/South-East Asia with an emphasis during El Niño and La Niña. *Atmospheric Research, 197*, 150-166.